# Three Alkaloids from an Apocynaceae Species, *Aspidosperma spruceanum* as Antileishmaniasis Agents by In Silico Demo-case Studies

**DOI:** 10.3390/plants9080983

**Published:** 2020-08-03

**Authors:** Diana Morales-Jadán, José Blanco-Salas, Trinidad Ruiz-Téllez, Francisco Centeno

**Affiliations:** 1Ecology and Earth Science, Department of Vegetal Biology, Faculty of Sciences, University of Extremadura, 06006 Badajoz, Spain; diana.moralesj91@gmail.com (D.M.-J.); truiz@unex.es (T.R.-T.); 2Molecular Biology and Genetics, Department of Biochemistry, Faculty of Sciences, University of Extremadura, 06006 Badajoz, Spain

**Keywords:** *Aspidosperma spruceanum*, aspidoalbine, aspidocarpine, tubotaiwine, in silico, folding, docking, Leishmania targets

## Abstract

This paper is focused on demonstrating with a real case that Ethnobotany added to Bioinformatics is a promising tool for new drugs search. It encourages *the* in silico investigation of “challua kaspi”, a medicinal kichwa Amazonian plant (*Aspidosperma spruceanum)* against a Neglected Tropical Disease, leishmaniasis. The illness affects over 150 million people especially in subtropical regions, there is no vaccination and conventional treatments are unsatisfactory. In attempts to find potent and safe inhibitors of its etiological agent, Leishmania, we recovered the published traditional knowledge on kichwa antimalarials and selected three *A. spruceanum* alkaloids, (aspidoalbine, aspidocarpine and tubotaiwine), to evaluate by molecular docking their activity upon five Leishmania targets: DHFR-TS, PTR1, PK, HGPRT and SQS enzymes. Our simulation results suggest that aspidoalbine interacts competitively with the five targets, with a greater affinity for the active site of PTR1 than some physiological ligands. Our virtual data also point to the demonstration of few side effects. The predicted binding free energy has a greater affinity to Leishmania proteins than to their homologous in humans (TS, DHR, PKLR, HGPRT and SQS), and there is no match with binding pockets of physiological importance. Keys for the in silico protocols applied are included in order to offer a standardized method replicable in other cases. Apocynaceae having ethnobotanical use can be virtually tested as molecular antileishmaniasis new drugs.

## 1. Introduction

### 1.1. Aspidosperma spruceanum, Indigenous Medicinal Plant

*Aspidosperma spruceanum* Benth. ex Müll. Arg is an endemic tree of the Apocynaceae family, with a neotropical and subtropical distribution, that lives in the ombrophilous rainforests of Central and South America from sea level up to 400 (−1000) m (Figure 1).

The species was firstly described in Flora Brasiliensis of Karl von Martius (1860), and has been subject to taxonomic treatments until relatively recent times [1] (Table 1)

It has a characteristic reddish latex, lenticellated cortex; and big coriaceous, oblong acute discolor leaves, craspedodromous nerved. The pentamerous yellow flowers are grouped in corymbiform inflorescences. As described by their monographs, tomentose calix (2–2.5 mm) of ovate-acute lobes (1.5–1.7 mm) are present; as well as a salviform corolla (6.5–7 mm) which is glabrous externally and tomentose internally below the stamen [1,2]. The androecium is tomentose; the gynoecium glabrous and the style-head has 2 oblong apical appendages. Its follicles (10–14 cm) are dolabriform. Orbicular seeds have 5.8–6.6 cm of diameter (Figure 2).

In Amazonian Ecuador, indigenous kichwa populations collected specimens of what they called “challua kaspi”. Ethnobotanical studies made under the clauses of the Nagoya Protocol [3] identified it as *A. spruceanum*, referring as well that it had been used in ancestral kichwa medicine against body ache and malaria [4].

In other Amazonian countries, Aspidosperma had also been used to treat malaria and leishmaniasis [4,5,6,7]. Both are parasitic illness caused by obligate intracellular protozoa, transmitted by insect bites. But they are different challenges to tackle from a pharmacological point of view. Malaria treatments are much more available. Leishmaniasis still the remains in the list of Neglected Tropical Diseases of the World Health Organization.

### 1.2. Computer Aided Drug Design Approaches Catalyzed by Ethnobotany

Leishmaniasis is a widespread disease for up to 88 countries around the world, most of them with low Human Development Index. It is produced by (up to 20) species of Leishmania with different geographical distribution and clinical manifestations patterns [8,9,10,11]. In tropical regions America, it is transmitted to humans by Lutzomyia. The treatment includes antimonial compounds, amphotericin B, paromomycin, pentamidine and miltefosine, but they induce resistance mechanisms, so new drugs are needed [12]. This requires an investment that it is not affordable if there is not a profitable market. This coincides with the socioeconomic conditions of the countries where the disease develops [13,14,15]. For this reasons Leishmaniasis is a NTDs [12]. The search for new bioactive compounds from plants studied by Ethnobotanists has frequently contributed to the discovery and development of many drugs with therapeutic applications [16]. Today, new drugs searching can be done by the computer-aided drug design (CADD) approaches [17]. It consist in the application of computer science to search for molecules capable of inhibiting activities of target proteins [17,18]. Thus, predicting the ligand–protein binding sites and the biochemical function of proteins offers a practical alternative solution and has become a most valuable preclinical method [19]. It can reduce the use of animals for in vivo testing, decrease the cost of drug discovery and speed the process up [20]. The bibliographic review of Aspidosperma from the phytochemical point of view will offer as a result a significant number of chemical structures as is usual in the Apocynaceae, mainly alkaloids. Alkaloids have been considered in literature as potential antileishmaniasis. All these molecules are available to be used randomly used in CAAD, as it is usual to be done with Chemical Libraries of Molecules uploaded to International Molecular Databases for Drug Design.

In this paper we suggest as an innovation to incorporate traditional knowledge. It will not avoid clinical trials. But it can speed up preclinical phase or lead us to surprising in silico results Descriptive Phytochemical published information, which is very abundant from the middle of the last century, can from nowadays on be better valued through CAAD and Ethnobotany [20].

*Aspidosperma spruceanum* is a plant used by Kichwas from Ecuador against malaria. In Ecuador there is a similar illness, leishmaniasis, which is a neglected tropical disease that hits severely the country, lacking of effective treatments. Sometimes Amazonian uses this plant against both diseases. Medicinal Chemistry let us nowadays to explore in the databases and work in bioinformatics. From this framework we have raised up the present approach which aim is to find out if some metabolites of *A. spruceanum* can be coherent with the antileishmaniasis properties attributed to this plant by the Amazonian traditional knowledge. We pretend to offer a methodology that supposes a competitive advantage in the preclinical phase of new drug trials.

Performing tests in silico to investigate the action of molecules of plants only can be made with botanical families that have been previously object of chemical composition analysis. This is the case of Apocynaceae. Many taxa of this family could serve to replicate protocols such as the one below described. We have selected *A. spruceanum* based in its indigenous use against Leishmania.

The specific objective of this paper is to investigate the action of metabolites from *Aspidosperma spruceanum*, in the active sites of some Amazonian Leishmania targets.

## 2. Results

### 2.1. Selection and 3D Representation of A. spruceanum Metabolites

*A. spruceanum* is a well-recognized taxon, where the literature mentions 19 different indole alkaloids [21,22] from which three were selected for the present In silico Demo-Case Study. They fulfill Lipinski’s rule, which, theoretically, is important for good drug absorption and permeation through biological membranes. These were: aspidocarpine, aspidoalbine and tubotaiwine (Figure 3, Table A1 and Appendix A in Appendix A).

### 2.2. Modelling Leishmania Targets and Docking with Physiological Ligands

The bibliographic prospection revealed that in Ecuador leishmaniasis is mainly produced by 4 species of Leishmania: *L. braziliensis*, *L. panamensis*, *L. amazonensis* and *L. mexicana,* and the most relevant targets for leishmaniasis expressed by them are: Dihydrofolate reductase-thymidylate synthase (DHFR-TS); Pteridine Reductase 1 (PTR1); Pyruvate kinase (PK); Hypoxanthine-guanine phosphoribosyltransferase (HGPRT); Cysteine proteases (CP); Superoxide dismutases (SOD); Inositol phosphorylceramide synthase (IPCS) and Squalene synthase (SQS) [25,26,27,28].

In the direct search for crystallized 3D structures of these targets, it was found only one 3D Ray Crystal structure for PK of *L. mexicana*. The rest of the targets have not an available crystallized structure scannable with X-Rays. For this reason, modeling was addressed, comprising prediction of 3D structure with aminoacid sequence from the chosen isoform. More than 20 were found, of which 8 were selected, one for each target, prioritizing the species with more available information, which turned out to be *L. panamensis*. Following our methodological proposal, modelling was performed, discarding those models without the established quality requirements (Figure A1 and Table A2). As a final result we obtained the 5 target models shown in Table 2.

The predicted binding free energy corresponding to the respective catalyzed reaction between the targets of Table 2 and the corresponding physiological ligands (Table 3) contains ΔG values that suggest good level of stability in the protein-ligand binding.

### 2.3. Efficacy Tests: A. spruceanum Metabolites vs. Leishmania Targets

Three indole alkaloids from *A. spruceanum* (APA, APC, TBT) were tested against the selected Leishmania targets, resulting predicted binding free energy values lower than the obtained with some physiological ligands (Table 4).

### 2.4. Ligand Binding Affinity Tests: A. spruceanum Metabolites vs. Human Targets

The docking results over *H. sapiens* homologue of Leishmania are shown in Table 5. Taking them in consideration we have highlighted in bold in Table 4 the most stable bindings by computational results.

### 2.5. Active Site Identification

#### 2.5.1. For In Silico Activity Testing

The binding pockets, identified either by literature review or by the position of physiological ligands (Table A4), were represented for better understanding in Appendix C (Figure A3, Figure A4 and Figure A5), next to the superposition of ligands. The cavity shape where all the docking calculations were done is incorporated in the Appendix A.

#### 2.5.2. For In Silico Ligand Binding Affinity Testing

Molecular docking superposition between Leishmania and *Homo sapiens* targets was made. The interactions with the residues located in the regions that are relevant to the selectivity towards physiological ligands were: for NADP (Leu18, Leu66 and Ser111), for THB (Gly13, Lys16, Arg17, Asn109, Lys198 and Tyr194), and for the lipophilic pocket with MTX (Lys16, Hsd38 and Gly225). If we compared the interaction of APA (Arg17), APC (Lys 198, Arg17) and TBT (Arg17 and Tyr194), whose predicted binding free energies were −8.85, −9.02 and −8.29 kcal/mol, respectively (Table 3, Table 4 and Table 5, Figure A2 and Figure A3), the three molecules were able to be accommodated in the binding site of PTR1, reaching binding free energies lower (stronger binding) than some physiological ligand/substrates shown in Table 3.

It can be observed that the pocket of Leishmania PTR1 possesses a greater volume than the human DHR and a different shape. Molecules with a bigger size (coenzymes) possess lower predicted energy values. Like APA, APC and TBT appeared to closely interact with the physiological ligands site (control) in the computational structure prediction of PTR1 (Figure A5).

The positions of these plant derived ligands on the Leishmania target suggest that the active sites of the homologous enzyme in humans will not be affected, corroborating this information with the positions and values of ΔG kcal/mol with more affinity in parasite targets.

These results allow us to identify the importance of the position of the three plant metabolites in the Leishmania PTR1 that partially overlap with the NADP site (Rossman motif). In addition, it points to how the residues involved in human DHR with plant-based compounds (green zone, section B) do not interfere in the binding pocket of interest (Figure 4).

## 3. Discussion

The debt that pharmacy and medicine owe to nature is of immeasurable value, and a part of this debt is due to Ethnobotany. The ethnobotanical knowledge of the Kichwas communities about the medicinal use of the *Aspidosperma spruceanum* led us to select three natural metabolites of this plant candidate to research the efficiency in the treatment of leishmaniosis. It was faced by application based drug design. In this scenario, the crystal structure and structural data for validated targets are the main requirements. If they lack, a longer protocol has to be implemented, facing to model the structure of the target proteins, with the help from available templates to suit each case. A previous study in a proximate species (*L. donovani*) had been carried out with was carried out in this framework, obtaining satisfactory results [13,34].

In our case we have applied a general protocol for the use of bioinformatics tools aimed at discriminating potential ligands for target proteins, although the structure of these are still unknown, which is summarized in Figure 5. Thus, in regard to the possible therapeutic targets of Leishmania, we obtained eight possible candidates through an exhaustive bibliographic search. As for the target proteins, after searching for its sequences, we verified if its structures were unknown. When the structure of the target protein was unknown, homology modelling was carried out with proteins of similar function and known structure. The structural models obtained were analyzed according to the parameters by protein structure homology-modeling server, which decreased the number of targets to five.

For the later stage of docking ligand protein, one of the limiting factors is that the structure of the target protein is well resolved. Since the structures obtained come from folding by homology, we decided to check the goodness of these models by doing docking with their physiological ligands. This allowed us to compare the affinity between each protein and its physiological ligands with respect to the affinity with other homologous proteins assigned to these ligands. The results obtained for the predicted free energies of the complexes were very similar, and in the same order of magnitude and consistent with the IC50 or Kd of the order of hundreds μM to fews μM. These values match well with the intracellular concentrations of the physiological ligands used. These results allowed us to conclude [29,35,36,37], that the three-dimensional structures of the five Leishmania targets were very suitable for the next stage: docking with the selected plant natural products.

On the other hand, we selected homologous proteins in *H. sapiens*, whose three-dimensional structures were resolved by X-ray diffraction, but we carried out a sequence comparison using BLAST, finding similarities ranging between 28% and 61%. We believe that the restrictions and controls imposed should allow us to dock with the *A. spruceanum* metabolites, as well as compare their binding in the Leishmania protein and in the human homologue to find out how specific is the plant alkaloid.

The molecular docking simulations results suggest that metabolite binds directly at the active site, except in PK. They are very likely to interfere with the function since they generate steric hindrance (DHFR-TS, PTR1, and SQS) or they bind in the same positions as physiological ligands (APA in SQS). The greater binding affinity, indicated by the lower docking score of −7 kcal/mol, propose the stronger inhibitory activity, especially of the three metabolites against the PTR1 target compared to the other enzymes. Computational results suggest that APA, with a predicted binding energy of −8.15 to −8.99 kcal/mol, is a metabolite with more affinity in Leishmania than in *H. sapiens.* These occupy places of interest, such as coenzyme at the NADP biding site and substrates. All of this reinforces the validity of the results of the performed docking tests.

The DHFR-TS target docking suggests the positions of plant metabolites similar to NADP, THF, and dUMP presenting predicted energy values of −9 kcal/mol, similar to that of the dUMP substrate (−10 kcal/mol) considered our control. In the superposition with human structures, only APA coincides in the Arg50 union with dUMP, but the union energy has less affinity with the human target (see Appendix A).

The docking in the PTR1 target predicts that most of the studies *A. spruceanum* metabolites and physiological ligands are in the same pocket. The THB substrate has a predicted binding energy close to the three metabolites, indicating a good prognosis of pharmacological activity. Positions of interest in humans are not affected. However, the binding sites of the three plant metabolites studied in the homologous human protein, DHR, are located far from the NADP binding site. This could explain the higher affinity found in the binding of these plant metabolites to the Leishmania target versus the *H. sapiens* target.

The PK enzyme docking concludes that no plant metabolite resembles any physiological ligand in position at binding energy values less than −8 kcal/mol but at higher values if, in which the metabolites compete with values similar to those of PVT (−7.6 kcal/mol), the stereochemistry and arrangement of the molecules have been considered. In the overlap with the target of humans, the plant metabolites in humans possible do not interfere in the positions of the substrates (see Appendix A).

The docking in the HGPRT target suggests that the physiological ligand coincides in position with the plant alkaloids, whose predicted free energy values are very close, between −8 and −9 kcal/mol, forming the residue Thr133 an H-bond with APA and with 5GP, and making the link strong (see Appendix A).

The SQS enzyme in its coupling predicts how the three *A. spruceanum* studied alkaloids share a position with NADPH and FPS. In the superposition with the human target, it is indicated that it does not affect the active site and that they share binding pockets with the two physiological ligands, point to that only APA has a greater affinity for Leishmania than for humans (see Appendix A).

The cavity detection method is directly associated with the druggability prediction; the application of this method can provide insights into the druggable targetome contained in the structural proteome. This is a useful approach to unlock promising yet largely unpursued mechanisms.

These bioinformatic studies reinforce the ancestral use by Amazonian kichwa of *A. spruceanum* against Leishmania. Therefore, future studies of these bioactive molecules or their analogs can be focused on in vitro, in vivo and clinical trials. In addition, this type of bioinformatic approach, with control strategies equal to or similar to those applied in this work, can be continued with other plant metabolites with an ethnobotanical background, which would allow us to find new lead compounds, and control the disease and as far as possible, generating minus adverse reactions in the host.

## 4. Materials and Methods

The Programs/Database used for the bioinformatic approach are listed in Table A5 (Appendix B).

### 4.1. Selection and 3D Representation of A. spruceanum Metabolites

The botanical synonyms of this species′ Latin name (Table 1) were searched in a specialized database: Tropicos. This is the most complete and up-to-date systematic source of tropical plants, especially Latin American ones. Each name was associated with the keywords: “chemical composition” or “phytoconstituents” or “phytochemistry”. Bibliographic search was made using Scopus, WOS and Google Scholar. Search results were filtered by selecting articles containing a study of the chemical composition of the chosen species, in addition to the name of the components in common nomenclature.

*Aspidosperma spruceanum* alkaloids with available structures in databases and some additional criteria were chosen (vg to be a starting structure, to have a similar structure of a pharmacophore). Corresponding IUPAC and ZINC codes were retrieved from Pubchem with MOL2 format.

Molecules were drawn using Marvin Sketch 19.15 converting the format of some structures into pdb files using OpenBabel version 2.3.1.

### 4.2. Modelling of Leishmania Targets; Physiological Ligands

A literature review was performed, in order to find:The most prevalent species of Leishmania in EcuadorThe most relevant and common targets described for leishmaniasis

It was made following a Prisma 2009 flow diagram methodology with the databases Scopus, Dialnet, Medline, PubMed, ScienceDirect, Google Patents, Google Scholar, and Wiley Online, using keywords as “Leishmania”, “Ecuador” and “target” [38].

The obtained results for this first step were:The scientific name of the infecting taxaThe capital letter acronyms (names) of the targets

Targets are proteins that have different isoforms in different species of Leishmania and in *Homo sapiens*. The names of the targets appear in bibliography as capital letters acronyms. 3D representation of targets is essential to understand the action mechanism of antileishmaniasis drugs. In order to find them we undertook a second step using the dichotomous key presented below, which is based in a previous proposal of our research group [39]. The process must be repeated for each of the species mentioned (=*taxon*).

Key for the in silico protocol applied:*Protocol N: .....*Target: ......*Taxon: ......

Go to www.rcsb.org (RCSB PDB, the Protein Data Bank of the Research Collaboratory for Structural Bioinformatics), insert the name of the target and the selected “Scientific Name/Source of Organism” (taxon).If 3D Ray Crystalline structures appear, go to .......................................................................................................................2If they do not appear, you must start modelling. Please go to .............................................................................................33D representation is obtained. The target is ready for docking, as described in below Section 4.4.Go to www.ncbi.nlm.nih.gov (NCBI, National Center for Biotechnology Information), select Protein (database). Insert the corresponding names in this format: *target taxon*, and search:If RefSeq (=predicted aminoacid reference sequences) code with format XP_NNN. N appear (X, prediction; P protein; N numbers; N isoform), go to ..................................................................................................................................................................... ...................................4If they do not appear, you must look for the nucleotide sequence, going to ........................................................................5**Select an isoform**. The sequence elected for each enzyme is the result of if any, of greater number of amino acids and the canonical or consensus sequence that it is the calculated order of most frequent residue.A predictive code XP_NNN.N, of target aminoacid sequence and selected isoform is obtained. XP_NNN.N, is ready to start modelling, as described in......................................................................................................................................................................................................6Go to www.ncbi.nlm.nih.gov and select Nucleotide (database), insert the name of *the target;* search and filter the results by taxon:If FASTA (=predicted nucleotide sequences) file is available, select it, copy, paste and save it and go to ..........................7If FASTA is not available the bioinformatic prospection will temporarily stop.Insert the XP_NNN.N in the www.ncbi.nlm.nih.gov and select the corresponding FASTA code; copy, save it and go to........................................7In www.swissmodel.expasy.org (Swiss-Model Program, for homology-modelling protein structures) insert the FASTA code, and select the option templates [40,41,42,43,44]. The result will be a group of 3D target models (SMTL, SwissModel Template) and we select the most reliable considering:

○Sequence identity >50%,○Templates: structures by X-ray crystallography with a resolution higher than 2.2 Å○GMQE (Global Model Quality Estimation) with higher numbers, better○QSQE (Quaternary Structure Quality Estimation) above 0.7, better○QMEAN Z 0 good value between model/experimental structures○High sequence similarity/Experimental resolution

The selected target model is saved with a six digit code: (SMTL-ID), the first 4 digits are the PDB ID.

Each target model may have 1 or more physiological ligands. The corresponding information for each case was retrieved from ligand list of the Model Building Report (Swiss-Model) and Uniprot (Function Section).

### 4.3. Docking Tests

We used SwissDock for the in silico simulation of protein–ligand docking [45,46]. We have considered that, of all those possible interactions, the most likely is the one with the lowest predicted free energy (ΔG), because it has the strongest binding affinity, and therefore would be the one whose grouping or cluster is most likely to spontaneously occur. The program provides us with ΔG values in kcal/mol allowing to download a group of 50–100 clusters. Each cluster can be identified by a C:E code number. The most stable reaction, has the lowest free energy value, and its corresponding C:E* number must be used as a locator.

Two series of tests docking leishmania targets were performed: one with physiological ligands and the other with the *A. spruceanum* metabolites. These were named activity tests because they point to the capability of the *A. spruceanum* molecules for modulating Leishmania action.

The next stage (named as ligand binding affinity tests) consisted in finding out if *A. spruceanum* metabolites interact with certain human proteins: the homologous to the leishmania targets.

Key for the in silico protocol applied

Go to www.swissdock.ch; enter the (SMTL-ID) target and (ZINC-AC) ligand or upload file, submit docking and go to ...........................................2Download the docking with C:E* clusters, identified the lowest ΔG and go to Section 4.5

### 4.4. Homo Sapiens Proteins Homologous to the Leishmania Targets 3D Representation

The EC number of Leishmania targets were searched in the Enzyme-Data-Base or Enzyme-of-Expasy, using this number en UniProt database. All enzymes were treated with Blast-p to confirm homologues with humans, represented with E-value [47,48].

Key for the in silico protocol applied

Go to www.enzyme-database.org/search.php; enter the name of Leishmania target, copy EC number and go to.....................................................2Open https://www.uniprot.org/ (Uniprot, resource of Protein sequence and functional information), paste EC number, select the Uniprot Code in accordance to the organism and copy the sequence and go to.....................................................................................................................................................3Open BLASTp https://blast.ncbi.nlm.nih.gov/Blast.cgi?PAGE=Proteins, paste sequence, select in organism optional *Homo sapiens* and write in expect threshold of algorithm parameters 0.00001.......................................................................................................................................................4Run BLAST and select accession of the homologues human..............................................................................................................................................5Go to www.rcsb.org (RCSB PDB), insert the name of *the target* and the selected “Scientific Name/Source of Organism” (*taxon*)............................Section 4.3

### 4.5. Active Site Identification

#### 4.5.1. For In Silico Activity Testing

The positions occupied by the plant metabolites in the leishmania targets (and human homologous ones) were compared with the information of the active sites of the physiological ligands and thus we estimated whether they (plant metabolite) could act as inhibitors of the abovementioned targets in both species *L. panamensis* and *H. sapiens*. We used Computed Atlas of Surface Topography of proteins server, CASTp 3.0, and the information was corroborated with the UniProt database. It was highlighted in the structure by using USCF Chimera 1.13.1 [49]. The binding pockets were colored by the hydrophobicity of the amino acids, ranging from blue for the most hydrophilic, white (neutral), orange and red for the most hydrophobic.

#### 4.5.2. For In Silico Ligand Binding Affinity Testing

The overlap the spatial structures of Leishmania target proteins and their human counterparts, was evaluated by superimposing them with the Program USCF Chimera. A bigger overlap points to estimate that the active binding sites of the plant metabolites have the same location in the respective homologous targets (leishmania and human), and this result must be estimated in terms of ligand binding affinity.

## 5. Conclusions

In this work, we studied the potential inhibitory activity of the plant based molecules APA, APC and TBT against the Leishmania targets DHFR-TS, PTR1, PK, HGPRT, and SQS, using bioinformatics tools. Our results show that the three plant metabolites are able to establish strong interactions with the targets, with predicted free energy values less than −7 kcal/mol, lowest (stronger) than some native ligands/substrates. It was concluded that APA, in all the targets, could inhibit some of the functional aspects of *L. panamensis* in particular, and thus may be useful for drug design studies. These findings contribute to the understanding of the effect of the bioactive metabolite on the parasite. Our studies provide important results for the biological potential of *A. spruceanum*, with promising pharmacological action to be investigated in the future as an alternative source for drug discovery, or for chemical modifications towards the synthesis of analogs with improved properties. These results can contribute to an increase in our knowledge of this species from the biodiversity of Ecuador.

Bioinformatics tools are useful for previous experimental research. We have shown that the plant-derived compounds of this study are very promising and effective as an antileishmanial agent. This information can provide important biochemical clues about the affinities of pharmacological objectives.

This in silico study opens borders, allows affordable and truthful measures and uses information available for free. Finally, the combination of these bioinformatics tools with ancestral ethnobotanical knowledge offers enormous advantages: it limits the number of molecules to be studied, values a natural product of a certain region, and allows accessible studies with a relatively low cost.

## Figures and Tables

**Figure 1 plants-09-00983-f001:**
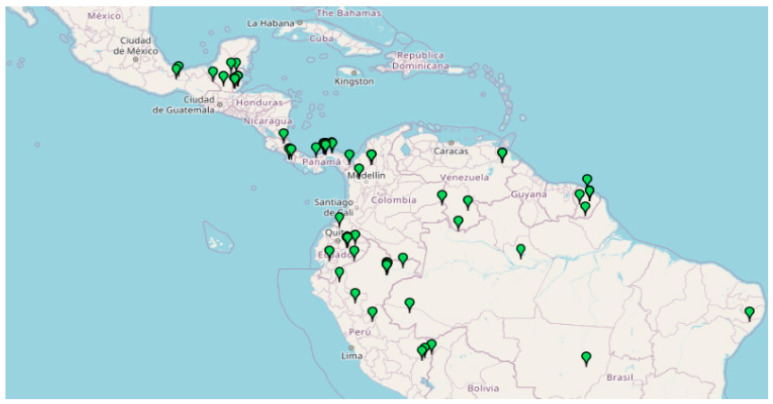
Distribution of *A. spruceanum* in Belize, Bolivia, Brazil, Colombia, Costa Rica, Ecuador, French Guiana, Guatemala, Guyana, Honduras, Mexico, Nicaragua, Panama, Perú, Suriname and Venezuela (retrieved from Missouri Botanical Garden, 2020).

**Figure 2 plants-09-00983-f002:**
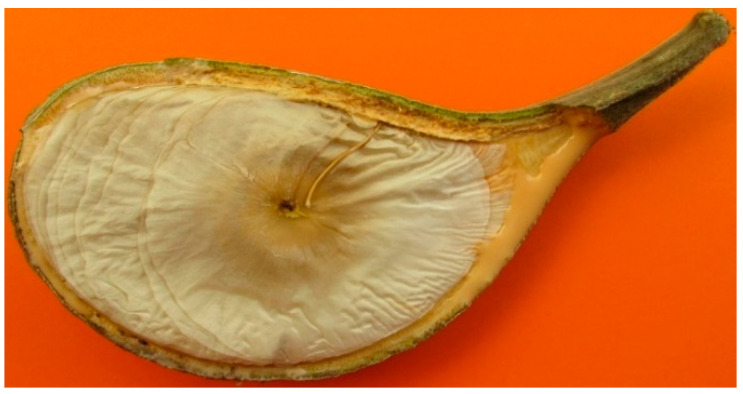
Orbicular seed of *Aspidosperma spruceanum* Benth. ex Müll.

**Figure 3 plants-09-00983-f003:**
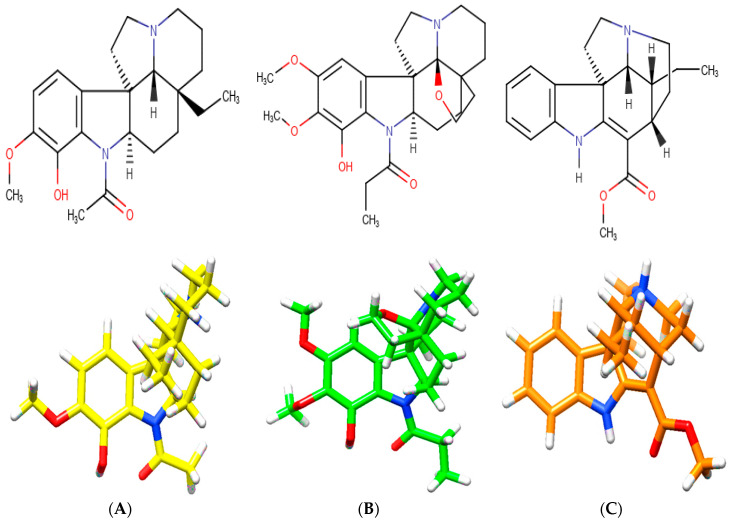
Chemical structures of indole alkaloids isolated from *A. spruceanum* and represented in 3D. (**A**) aspidocarpine (APC) [23,24]; (**B**) aspidoalbine (APA); (**C**) tubotaiwine (TBT) in reference pH. The heteroatoms are nitrogen (**blue**), oxygen (**red**) and hydrogen (**white**).

**Figure 4 plants-09-00983-f004:**
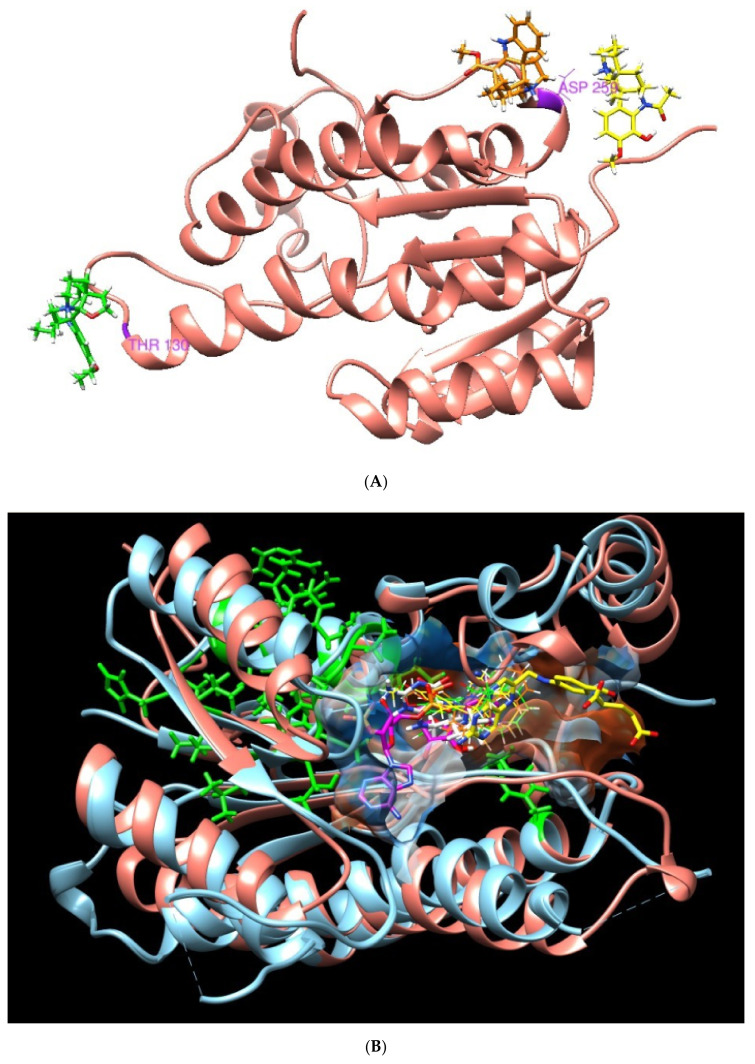
3D molecular structure of human DHR and Leishmania PTR1. (**A**) Position of metabolites APC (**yellow**), APA (**green**) and TBT (**orange**) in DHR human target. (**B**) Superposition of active sites. Large green areas represent positions of physiological ligand binding sites.

**Figure 5 plants-09-00983-f005:**
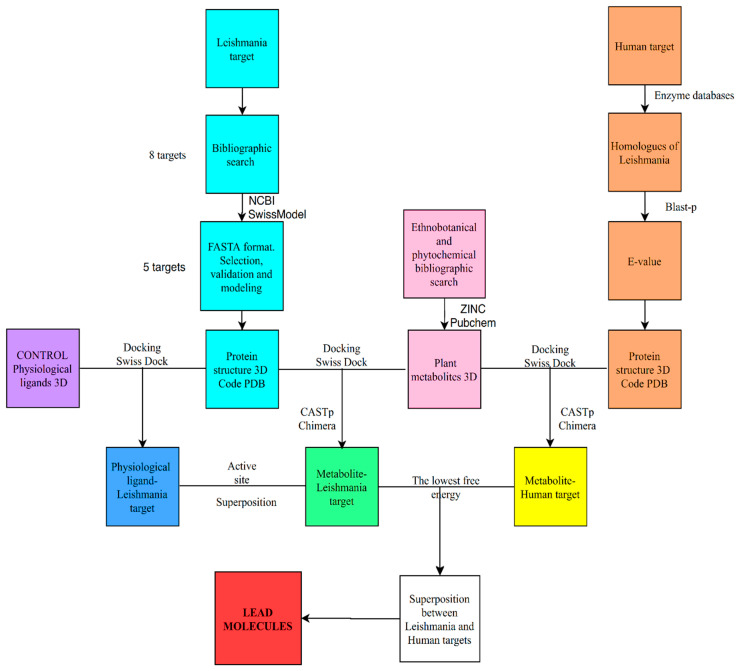
Flow chart of the in silico methodology followed in this work.

**Table 1 plants-09-00983-t001:** Synonyms of *Aspidosperma spruceanum* Benth. ex Müll. Arg. Retrieved from Tropicos.

Synonyms
*Aspidosperma chiapense fo. tenax* Matuda
*Aspidosperma desmanthum* Benth. ex Müll. Arg.
*Aspidosperma megalocarpon* Müll. Arg.
*Aspidosperma cruentum* Woodson
*Aspidosperma igapoanum* Markgr.
*Aspidosperma matudae* Lundell
*Aspidosperma steinbachii* Markgr.
*Aspidosperma melanocalyx* Müll. Arg.
*Aspidosperma woodsonianum* Markgr.
*Aspidosperma album* (Vahl) Benoist ex Pichon
*Macaglia spruceana* (Benth. ex Müll. Arg.) Kuntze

**Table 2 plants-09-00983-t002:** Studied targets and modelling code references.

Target	SMTL-ID	Code RefSeq	Modelling Target		
			PDB ID	Organism	Reference
Dihydrofolate reductase-thymidylate synthase (DHFR-TS)	3inv.1.A	XP_010703963.1	3INV	*T. cruzi*	[29]
Pteridine Reductase 1 (PTR1)	1e7w.1.A	XP_010699188.1	1E7W	*L. major*	[30]
Pyruvate kinase (PK)	3hqn.1.A	XP_010702429.1	3HQN	*L. mexicana*	[31]
Hypoxanthine-guanine phosphoribosyltransferase (HGPRT)	1pzm.1.B	XP_010698883.1	1PZM	*L. tarentolae*	[32]
Squalene synthase (SQS)	3wca.1.A	XP_010701665.1	3WCA	*T. cruzi*	[33]

**Table 3 plants-09-00983-t003:** Predicted binding free energy and hydrogen bonds. The physiological ligands were docked in five Leishmania targets according to the reaction catalyzed.

Physiological Ligands	$\;∆G\;\left(\frac{kcal}{mol}\right);\;\mathrm{Hbonds}$				
	DHFR-TS	PTR1	PK	HGPRT	SQS
NADP	−14.37; 8	−15.51; 11			
dUMP	−10.94; 5				
THF	−12.49; 10				
MTX		−12.13; 3			
THB		−7.63; 9			
FDP			−13.24; 5		
PVT			−7.60; 2		
ATP			−15.17; 3		
5GP				−9.89; 8	
FPS					−13.14; 5
NADPH					−16.07; 9

**Table 4 plants-09-00983-t004:** Predicted binding free energy and hydrogen bonds. *A. spruceanum* metabolites were docked in the selected Leishmania targets. Reference, 7.4 or/and 8.4 indicate pH of the medium.

	$\;∆G\;\left(\frac{kcal}{mol}\right);Hbonds$						
	APA		APC			TBT	
**Targets**	**Reference**	**7.4**	**Reference**	**7.4**	**8.4**	**Reference**	**7.4**
DHFR-TS	−8.17;2	−8.99; 1	−9.33; 3	−8.61; 1	−9.57; 1	−9.26; 1	−8.45; 1
PTR1	−8.01; 1	−8.85; 1	−8,36; 1	−9.02; 2	−9.66; 2	−8.29; 2	−8.17; 1
PK	−8.15; 0	−8.52; 0	−8.57; 0	−8.65; 1	−8.45; 3	−9.01; 2	−8.78; 1
HGPRT L.	−6.77; 0	−8.39; 1	−8.25; 2	−8.48; 0	−8.4; 1	−7.98; 1	−7.66; 0
SQS L.	−8.73; 1	−7.98; 0	−7.41; 1	−7.53; 0	−9.31;2	−7.25; 1	−7.36; 0

**Table 5 plants-09-00983-t005:** Predicted binding free energy and hydrogen bonds. The metabolite molecules were docked in five *Homo sapiens* targets (additional data in Table A3). Reference, 7.4 or/and 8.4 indicate pH of the medium.

	$\;∆G\;\left(\frac{kcal}{mol}\right);Hbonds$						
	APA		APC			TBT	
**Targets**	**Reference**	**7.4**	**Reference**	**7.4**	**8.4**	**Reference**	**7.4**
TS	−7.88; 3	−8.33; 2	−8.9; 0	−8.62; 1	−10.06; 3	−9.15; 2	−8.40; 0
DHR	−8.25; 2	−6.79; 0	−7.49; 1	−7.03; 1	−8.94; 1	−6.83; 0	−6.79; 0
PKLR	−7.48; 0	−8.34; 2	−7.81; 0	−8.51; 1	−9.50; 1	−7.81; 0	−8.13; 0
HGPRT	−6.61; 0	−7.34; 0	−7.02; 0	−8.07; 1	−7.28; 0	−7.24; 0	−6.57; 0
SQS	−8.53; 0	−9.36; 1	−10.41; 1	−8.84; 0	−9.49; 0	−8.98; 1	−8.97; 0

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
