# Peer review of "Three Alkaloids from an Apocynaceae Species, Aspidosperma spruceanum as Antileishmaniasis Agents by In Silico Demo-case Studies"

_plants, 2020, doi:10.3390/plants9080983_

Round 1

Reviewer 1 Report

This manuscript describes mainly on the evaluation of three alkaloids from an Apocyanceae species, Aspidosperma spruceanum as antileishmaniasis agents by in silico study. Since it deals with original and interesting results, and I am recommending this article for publication in the Plants after the following issues have been addressed. The issues to be addressed include:

i) Overall the manuscript, it is necessary to check English grammar or syntax error. It is required to get English-editing. In addition, the References Section needs correction according to the Author’s guidelines (Especially, italic or capital or lower case letters… Ref. 5,9,11,13,15,16,22,23,24,34,36 and 39; Page number is mission in Ref 44). The description of “in silico” should be written in italic letters throughout the manuscript.

ii) It is preferable to change the title into an example “Three Alkaloids from an Apocyanceae Species, Aspidosperma spruceanum as Antileishmaniasis Agents by in silico Demo-case Studies” to clarify the purpose of the paper. It is not necessary to mention the authority of the botanical name in this title. It is adequate to describe it once in the Introduction or Materials and Methods Section.

iii)  In Table 1, it is necessary to add other synonyms regarding to the data on theplantlists.org.

iv) In Figure 1, it is required to redraw the chemical structures according to ACS settings.

v) It seems not to be an adequate description about the result of toxicity test: can only the selectivity of the ligands toward Leishmania and human targets ensure lower toxicity of the candidate as therapeutic agents? It is preferable to change the term or to add other evidences (including references).

Author Response

This manuscript describes mainly on the evaluation of three alkaloids from an Apocyanceae species, Aspidosperma spruceanum as antileishmaniasis agents by in silico study. Since it deals with original and interesting results, and I am recommending this article for publication in the Plants after the following issues have been addressed. The issues to be addressed include:

i) Overall the manuscript, it is necessary to check English grammar or syntax error. It is required to get English-editing.

  • Thanks for the suggestion. We have requested from the the use of the MDPI Edition Services and this amendment will be made.

In addition, the References Section needs correction according to the Author’s guidelines (Especially, italic or capital or lower case letters… Ref. 5,9,11,13,15,16,22,23,24,34,36 and 39; Page number is mission in Ref 44).

  • The references have been modified with the Mendeley Bibliographic Manager using Plants style. The page number of Ref. 44 has been included in Line 602

The description of “in silico” should be written in italic letters throughout the manuscript.

  1. It has been modified, see Lines 4, 17, 29, 33, 89, 100, 113, 158, 163, 211, 306,356, 369, 378, 392, 401, 424.

ii) It is preferable to change the title into an example “Three Alkaloids from an Apocyanceae Species, Aspidosperma spruceanum as Antileishmaniasis Agents by in silico Demo-case Studies” to clarify the purpose of the paper. It is not necessary to mention the authority of the botanical name in this title. It is adequate to describe it once in the Introduction or Materials and Methods Section

  • We have considered your recommendation and changed the title, which is much better than the original one. Thank you very much. According to your suggestion we have deleted it. We have included it once in the Introduction, see Line 36

iii)  In Table 1, it is necessary to add other synonyms regarding to the data on theplantlists.org.

  • We have indirectly included them because the database we have used Tropics, is based on theplantlists.org among other data sources.

iv) In Figure 1, it is required to redraw the chemical structures according to ACS settings.

  • We have drawn the figures with Chem-Draw ACS-1996 document setting, but we do not know if there has been any trouble with the saving process. We are going to ask the MDPI Editing Service to help.

v) It seems not to be an adequate description about the result of toxicity test: can only the selectivity of the ligands toward Leishmania and human targets ensure lower toxicity of the candidate as therapeutic agents? It is preferable to change the term or to add other evidences (including references).

  • Thank you for the comment, because it really seems preferable to change the term. We have used instead the term ligand binding affinity as you can see in Lines 149, 163, 366, 401, 406.

Reviewer 2 Report

The manuscript persues an interesting approach, however there are
several shortcommings that must be resolved:

Major issues:

(1) The obtained binding energies are predicted values from docking
and scoring programs and not experimental result.
Therefore each corresponding mentioning throughout the manuscript
must indicated appropriately (e.g. "predicted free energy of binding")
Corresponding values are subject to substantial errors and uncertainties
that often makes it impossible to even distinguish between high and low
affinity binders.
Therefore the differences listed in Tables 3-5 do not point towards
a preference in any direction.

Likewise docking into homology models can lead to quite adventurous
results, particularly if there is no possibility to compare them with
experimental data of similar ligands.
Therefore the (basically all) results of this study must be treated
with caution.
Thus I strongly recommend to (re)formulate the corresponding
statements (particularly in the results and conclusion) more
carefully (e.g. "computational results suggest/indicate/point to").

(2) It does not make sense to list hydrogen-bond energies, only the
total energy is of interest. Tables 3-5
Moreover, at least one hydrogen-bond between protein and ligand
is required for selective binding. The absence indicates basically
useless docking conformations.
What is the meaning of "Reference" in Tables 3-5 ?
If it is the human enzyme refer to it as "human".

Minor issues:
Add "NTD" to the list of Abbreviations

page 3, running line 66: Please specify "America" more precisely in terms of geographical
location (Northern, Middle, Southern, throughout, tropical regions, etc.).

Figure 4: Mention in the legend the special function of APA (green) otherwise
the reader assumes that it binds as unspecific inhibitor.

Figure B2: side chains that from interactions should be visualized
as bonds and sticks. Alternatively 2D depictions of the interactions
are more helpful (like those on the PDB website).
The same applies for the corresponding figures in the supplementary material.

Figure B3 is too busy and does not provide substantial information.
Delete.

Author Response

The manuscript persues an interesting approach, however there are several shortcommings that must be resolved: Major issues:(1) The obtained binding energies are predicted values from docking and scoring programs and not experimental result. Therefore each corresponding mentioning throughout the manuscript must indicated appropriately (e.g. "predicted free energy of binding")

  • We have corrected this expression, and “predicted free energy of binding” changes have been introduced. Please see lines 27, 138, 141, 145, 147, 153, 169, 175, 217, 234, 238, 243, 256, 321, 322, 357, 411.

Corresponding values are subject to substantial errors and uncertainties that often makes it impossible to even distinguish between high and low affinity binders. Therefore the differences listed in Tables 3-5 do not point towards a preference in any direction. Likewise docking into homology models can lead to quite adventurous results, particularly if there is no possibility to compare them with experimental data of similar ligands.Therefore the (basically all) results of this study must be treated with caution. Thus I strongly recommend to (re)formulate the corresponding statements (particularly in the results and conclusion) more carefully (e.g. "computational results suggest/indicate/point to").

  • One of the objectives of this work is to validate the use of bioinformatics tools to select some molecules of plant origin from thousands as potential therapeutic agents. The bioinformatics tool offers many advantages, and one of them is its much lower cost compared to in vitro, ex vivo or in vivo work. Therefore, making an initial screening using bioinformatics techniques is an advantage because of the enormous savings it entails, the continuation in later works goes through in vitro studies to contrast the in silico results of the preselected molecules and their targets. We reformulated the corresponding statements in lines 140, 147-148, 169, 175,,182, 232, 233, 234, 238, 243, 253, 256, 262 in the direction of your suggestion.

(2) It does not make sense to list hydrogen-bond energies, only the total energy is of interest. Tables 3-5 Moreover, at least one hydrogen-bond between protein and ligand is required for selective binding. The absence indicates basically useless docking conformations.

  • The reason of including the hydrogen-bond energies is because the reader will understand more quickly and clearly the binding relationship between target and ligand, without the need to observe in 3D. Comparing mostly affinity to alkaloids

What is the meaning of "Reference" in Tables 3-5 ? If it is the human enzyme refer to it as "human".

  • The expression Reference, 7.4 or/and 8.4 indicate pH of the medium, as we have included in lines 148, 154, 155.

Minor issues:

Add "NTD" to the list of Abbreviations

  • It has been added in Line 443

page 3, running line 66: Please specify "America" more precisely in terms of geographical location (Northern, Middle, Southern, throughout, tropical regions, etc.).

  • It has been modified in line 70

Figure 4: Mention in the legend the special function of APA (green) otherwise the reader assumes that it binds as unspecific inhibitor.

  • The legend has been modified as you can see in line 189. These words have been introduced: “Large green areas represent positions of physiological ligand binding sites”

Figure B2: side chains that from interactions should be visualized as bonds and sticks. Alternatively 2D depictions of the interactions are more helpful (like those on the PDB website). The same applies for the corresponding figures in the supplementary material.

  • We have used this representation system because we think it is the best way to visualize the specific binding with the ligands, being able to additionally see the stereochemistry of the compound. Anyway we are going to consult the Editing MDPI Service in order to improve the quality and size of our visualizations, mainly in the supplementary material, in order to let downloading of images increasing the figure size.

Figure B3 is too busy and does not provide substantial information.Delete.

  • We decided to superimpose both human and Leishmania to indicate that the binding sites do or do not coincide with that of the physiological ligands, depending on the enzyme. We believe that it allows clarifying all the data, the same applies for the corresponding figures in the supplementary material.

Thank you for your time and consideration of our work.

Best regards,

José Blanco Salas & Francisco Centeno

Round 2

Reviewer 2 Report

The manuscript is now suitable for publication.